

# Entanglement and interaction in a topological quantum walk

**Alberto D. Verga[1]⋆, Ricardo Gabriel Elías[2]**

**1** Aix-Marseille Université, CPT, Campus de Luminy, case 907, 13288 Marseille, France
**2** Departamento de Física and CEDENNA, Universidad de Santiago de Chile,
Avda. Ecuador 3493, Santiago Chile.

⋆ alberto.verga@univ-amu.fr

## Abstract

We study the quantum walk of two interacting particles on a line with an interface separating two topologically distinct regions. The interaction induces a localization-delocalization transition of the edge state at the interface. We characterize the transition through the entanglement between the two particles.

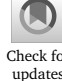

## 1 Introduction

Quantum walks are not only important in quantum information [1,2] as universal algorithms, [3,4] but also in condensed matter [5,6], as models of topological phases [7,8]. Quantum walks can also provide a simulation tool of quantum systems [9,10] or be used as an instrument to probe complex systems [11–13]. Various experimental implementations demonstrated their feasibility using ions or photons, including various walkers and interactions [14–18].

Topological nontrivial phases of matter are an active field of investigation since the discovery of the quantum Hall effect [19]. These condensed phases can emerge in widely different materials, from topological insulators [20,21] to chiral magnetic films [22,23]. A variety of

topological effects arise from the interaction between spin and momentum, as in the spin-orbit coupling: edge states in topological insulators and skyrmion lattices in magnetic materials arise from this coupling. It is interesting to note that the quantized anomalous Hall effect was experimentally exhibited in a three dimensional topological insulator [24], doped with magnetic impurities. Yet, quantum walks are based on the same kind of coupling. Indeed, in a quantum walk a particle at a site moves to the neighboring sites according to its spin state; in the simplest one dimensional walk the spin up projection moves to the right and the spin down projection to the left: after one walk step the particle is in a superposition of two sites and two spin projections. Therefore, it is a natural idea to use quantum walks from quantum information theory to explore topological effects from condensed matter physics [25].

In this paper we investigate the motion of two interacting walkers on a one dimensional lattice [26–29]. The lattice is partitioned into two regions with different topology, in such a way that an edge channel appears at the interface (in one dimension it reduces to a localized state at the origin) [30, 31]. One question to be asked is how does the interaction modify the edge state, or equivalently, what is the interplay between interaction and topology. We will see that the interaction can indeed create a bound state at the interface, when without interaction there is no edge state, and inversely, it can destroy an existing edge state. Another question that naturally arises is whether the existence of localized states modifies the behavior of the entanglement entropy. We will show that this is indeed the case, the entanglement growths at different rates depending on the way particles propagate.

After a detailed presentation of the model, we describe the phenomenology of the two particle quantum walk, then, we present our results on its localization and entanglement properties, followed by a brief discussion and a conclusion.

## 2 Model

The Hilbert space of a walker is defined by two quantum numbers, its position $x = -L/2, \ldots, L/2$ that can take $L$ integer values, and its spin $s = 0, 1$ (up and down, respectively). The Hilbert space of the two particles is the Kronecker product of the one-particle Hilbert spaces $\mathcal{H} = \mathcal{H}_1 \otimes \mathcal{H}_2$,

$$|x_1 s_1 x_2 s_2\rangle = |x_1 s_1\rangle \otimes |x_2 s_2\rangle \in \mathcal{H}. \tag{1}$$

The dimension of the two particle space is $\dim \mathcal{H} = (2L)^2$.

The spin state can be modified at each site by the unitary operator,

$$R(\theta) = 1_x \otimes \exp(i\sigma_y \theta), \tag{2}$$

where $1_x$ is the unit in the position Hilbert subspace, and the rotation angle $\theta \in (-\pi, \pi)$, could in principle depend on the site ($\sigma_y$ is a Pauli matrix). Each particle moves from site $x$ to a neighboring site $x \pm 1$, depending on its spin polarization; the operators $T_\pm$ shift the up or down spins to the right or the left, respectively,

$$T_+ = \sum_x (|x+1\rangle \langle x| \otimes |0\rangle \langle 0| + |x\rangle \langle x| \otimes |1\rangle \langle 1|), \tag{3}$$

$$T_- = \sum_x (|x\rangle \langle x| \otimes |0\rangle \langle 0| + |x-1\rangle \langle x| \otimes |1\rangle \langle 1|), \tag{4}$$

which are also unitary (we take the lattice spacing as the unit length). One step of the two particle quantum walk is defined by the unitary "evolution" operator,

$$U = V(U_0 \otimes U_0), \quad U_0 = T_- R(\theta_-) T_+ R(\theta_+), \tag{5}$$

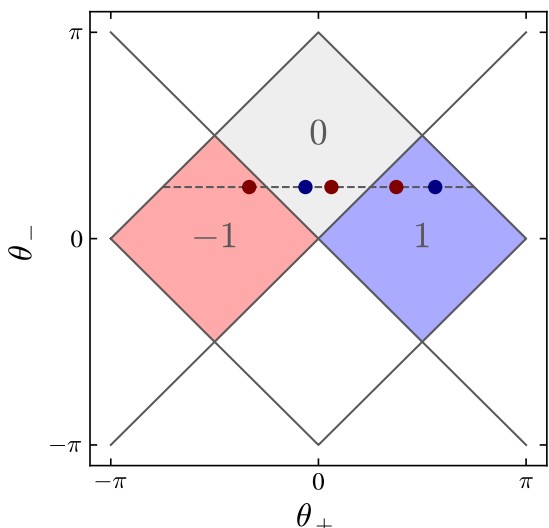

Figure 1: Topological index (charge) as a function of $(\theta_+, \theta_-)$; dots are for typical numerical parameters used: $\theta_- = \pi/4$ (dashed line), $\theta_+ = -\pi/3, \pi/16, 3\pi/8$ (for $x > 0$, red points), sweeping regions of charge $c = -1, 0, 1$, respectively, and $\theta_+ = -\pi/16, 9\pi/16$ (for $x < 0$, blue points), corresponding to the charges $c = 0, 1$. The solid lines separates parameter regions with different charge.

where $U_0$ is the one-particle operator and $V$ is the interaction operator,

$$V = 1 + \sum_{xs_1s_2}(e^{i\phi} - 1)|xs_1xs_2\rangle\langle xs_1xs_2|, \qquad (6)$$

where 1 is the unit matrix in $\mathcal{H}$ (the $-1$ term cancels with the first term when $x_1 = x_2$). The interaction acts only when the two particles share the same site, by adding a phase $\phi$ to the corresponding amplitude. The quantum walk is then characterized by the set of angles $\theta_\pm$ and the interaction phase $\phi$. One step of the walk is,

$$|\psi(t+1)\rangle = U|\psi(t)\rangle \qquad (7)$$

(we take the time step as the unit of time, and $\hbar = 1$) where $|\psi(t)\rangle$ is the quantum state of the walk at time $t$ (a vector of dimension $(2L)^2$). The dimension of the matrix $U$ is $(2L)^{2\times 2}$.

The split-step quantum walk defined by $U_0$ is known to possess nontrivial topology, depending on the choice of the pair $(\theta_+, \theta_-)$ [32]. In Fig. 1 we represented the topological phases of the split-step walk in the angles plane, and a set of typical parameters used in the numerical calculations. To study the interplay of interaction and topology in the quantum walk we introduce an interface separating two regions with different values of the topological charge $c$. These regions are defined by different choices of the rotation angles for the left ($\theta_L$) and right ($\theta_R = \theta$) parts of the interface:

$$\theta_+(x) = \theta_L + (\theta - \theta_L)\text{H}(x), \quad \theta_- = \pi/4, \qquad (8)$$

where $\theta_L$ can be $-\pi/16$ to set the left region charge to $c = 0$ or $9\pi/16$ to set it to $c = 1$ (as shown in Fig. 1 by the blue points), and we denote these two cases with the label 'c' ($c = 0, 1$); for each $\theta_L$ we vary the right angle $\theta = (-\pi/3, \pi/16, 3\pi/8)$, to account for $c = -1, 0, 1$ (the three black points of Fig. 1), respectively (H is the Heaviside function). We label 'i' ($i = 0, 1, 2$) the choice of $\theta$ for the three charges $c = -1, 0, 1$, respectively. We fixed $\theta_- = \pi/4$. Combinations of the labels 'c' (left charge) and 'i' (right charge) will be used to identify the parameters used in the calculations.

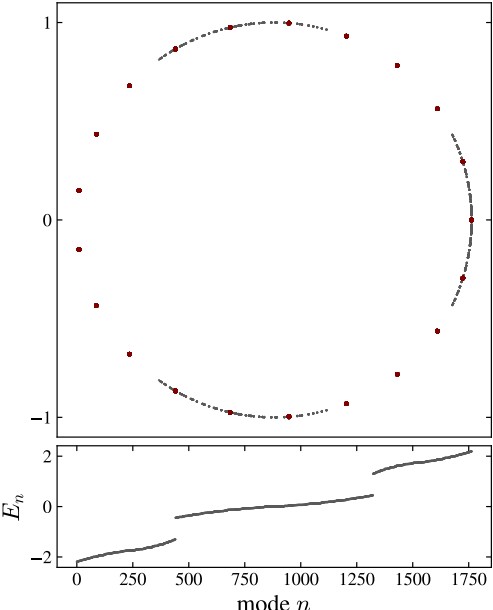

Figure 2: Quasienergies in the non interacting case, $\phi = 0$ (black points). (top) Distribution on the unit circle for $(\theta_+, \theta_-) = (3\pi/7, 2\pi/9)$. (bottom) The same quasienergies in increasing order, showing the band gaps. Red points are the eigenvalues of the quasimomentum $P$, whose eigenvectors are common to $U$, for a system size of $L = 21$. The gap arround $E = 0$ in the single particle system, splits into two gaps, as a consequence of the band wrapping characteristic of levels defined modulo $2\pi$.

Initial states are chosen to be Bell states with $x_1, x_2 = 0$,

$$|\phi_{\pm}\rangle = \frac{1}{\sqrt{2}}\big(|0000\rangle \pm |0101\rangle\big), \tag{9}$$

$$|\psi_{\pm}\rangle = \frac{1}{\sqrt{2}}\big(|0001\rangle \pm |0100\rangle\big) \tag{10}$$

or a separable state $|\psi_0\rangle = |0000\rangle$, where the +-sign states are symmetric and −-sign states are antisymmetric. We assign a label $b$ to these states which takes the values '0,1,2,3', for the Bell states, and '4' for the product state $|\psi_0\rangle$. Typical values for the interaction are,

$$\phi = (0, \pi/3, \pi/2, 3\pi/4, \pi),$$

labeled 'g' ($g = 0, \ldots, 4$), respectively. Therefore, the code

$$(\text{cbgi}), \ c = 0, 1, \ b = 0, \ldots, 4, \ g = 0, \ldots, 4, \ i = 0, 1, 2,$$

specifies the set of parameters used in the numerical computations: for instance (0321) corresponds to (left charge 0, state $\psi_-$, strength $\phi = \pi/2$, right charge $c = 0$).

In summary, the model consists in two particles moving in a one dimensional lattice with an interface at the origin; their evolution is ensured by a one time step operator $U$, composed of a coin (defined with different rotation angles in the two regions), which couples to the shift operator for each particle, and a local on-site interaction between the two particles. Although the interaction is spin independent, the global unitary operator cannot be splitted into a spin dependent part and a position dependent part. Coin and shift entangle spin and position in the one particle sector, and the interaction the positions in the two particle sector; as a

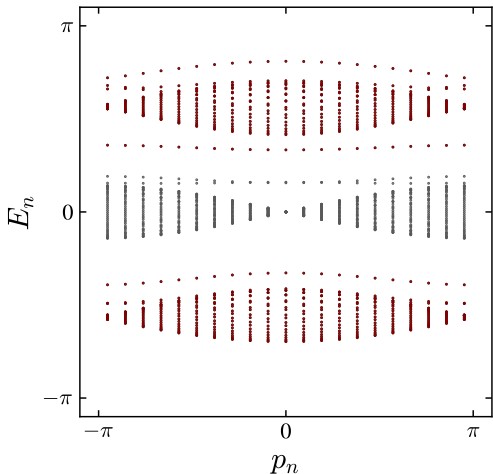

Figure 3: Band structure in the $(p_n, E_n)$ plane, for $\phi = \pi/3$ (other parameters as in Fig. 2). Note the appearance of bound modes within the gaps in the presence of interaction. These modes correspond to the the two particles occupying the same location. (The bands are colored for clarity.)

consequence, the interaction can modify differently the evolution of a given initial condition depending on its symmetry: symmetric and antisymmetric Bell states (9). In other words, $U = V (U_0 \otimes U_0)$ entangles spin and position ($U_0$) of each particle and the two particle positions ($V$), resulting in entanglement of all degrees of freedom labeled by the basis vectors $|x_1 s_1 x_2 s_2\rangle$ of the Hilbert space.

## 3 Results

The spectrum of the evolution operator is given by the solution of the eigensystem,

$$U |E\rangle = e^{-iE} |E\rangle , \tag{11}$$

with $E$ the quasienergies and $|E\rangle$ the eigenvectors. Since the interaction depends only on the distance between the two particles, the total quasimomentum $P = k_1 + k_2$, is conserved; here $(k_1, k_2)$ are the common eigenvalues of the translation operators $T_\pm$, for the two particles. Therefore, the energy eigenvectors $|E\rangle$, can be classified according to the eigenvalues $p_n$ of $P$ (see Figs. 2 and 3), which are the roots of the unity. In Fig. 2 we plot the quasienergies on the unit circle for the free system; the spectrum for the interacting system is shown in Fig. 3. We observe, in the interacting case, the appearance of bound states in the gaps, which correspond to the binding of the two particles (molecular state). This effect is similar to the Hadamard coin case investigated by Ahlbrecht et al. Ref. [27], and to a related model of two interacting fermions studied by Bisio et al. Ref. [29]. It is worth noting that, due to the wrapping of the energy bands (energies are defined modulo $2\pi$), the position and width of the gaps of the composite system do not trivially follow the one particle system, even for vanishing interaction strength. As a consequence, the parameter range for the existence of edge states (localized states at the origin $x = 0$), change for the two particles quantum walk.

To study how do the edge states depend upon the quantum walk parameters, we consider the eigenstates $|E\rangle$ of $U$. Let us start with the one particle case. The edge states should correspond to the $E = 0, \pi$ eigenmodes (Dirac cones). We are interested in the behavior of the wavefunction $\psi_E(x) = \langle x|E\rangle$, far form the interface $x = 0$. For a localized mode, we expect

an exponentially decreasing function of $x$,

$$\psi_E(x) \sim e^{-x/\xi_E}, \tag{12}$$

with a characteristic scale $\xi_E = \xi_E(\theta_+, \theta_-)$, the localization length of the eigenmode of energy $E$. To determine the localization length it is convenient to use the transfer matrix method: [33]

$$\xi_E^{-1} = \lim_{x \to \infty} \frac{1}{x} \log \left| \text{Tr} \prod_{n=1}^{x} M_n(E) \right|, \tag{13}$$

where $M_n$, the transfer matrix, relates the state at points $(x, x-1)$ to points $(x+1, x)$. The general case, with two particles, is not amenable analytically, however, we can consider the special case where the two particles share the same position, and restrict the evolution operator to the subspace $x_1 = x_2$:

$$U \to e^{i\phi} U_0,$$

leading to an effective one particle split-step walk with a phase change at each step. In this case the transfer matrix reduces to a two blocks $4 \times 4$ matrix $M$ independent of $x$:

$$\begin{pmatrix} \psi_0(x+1) \\ \psi_1(x+1) \\ \psi_0(x) \\ \psi_1(x) \end{pmatrix} = M \begin{pmatrix} \psi_0(x) \\ \psi_1(x) \\ \psi_0(x-1) \\ \psi_1(x-1) \end{pmatrix} \tag{14}$$

that can be written as

$$M = \begin{pmatrix} m & 0 \\ 0 & m \end{pmatrix}, \tag{15}$$

where, using $c_\pm = \cos\theta_\pm$ and $s_\pm = \sin\theta_\pm$,

$$m = \frac{1}{c_+ c_-} \begin{pmatrix} m_{00} & m_{01} \\ m_{10} & m_{11} \end{pmatrix}, \tag{16}$$

with

$$m_{00} = e^{-i(E+\phi)} + 2e^{i\phi} s_+ s_- + e^{i(E+2\phi)} s_+^2$$
$$m_{01} = e^{i(E+\phi)} c_+ s_+ + c_+ s_- = m_{10}$$
$$m_{11} = e^{i(E+\phi)} c_+^2 .$$

The eigenvalues $\lambda_\pm$ of $m$ give, from Eq. (13), the explicit expression of the localization length,

$$\xi_E^{-1} = \log \left| \max[\lambda_+(E), \lambda_-(E)] \right| = \Lambda(E). \tag{17}$$

Because the determinant $\det(M) = \det(m) = 1$, we have $\lambda_+ \lambda_- = 1$. For a related computation see Rakovszky and Asboth work on the localization in the split-step quantum walk Ref. [34]. For $E = 0$, we obtain,

$$\lambda_\pm(0) = \frac{e^{-i\phi}}{2c_+ c_-} \left[ e^{2i\phi} + 2e^{i\phi} s_+ s_- + 1 \pm \sqrt{\left( e^{2i\phi} + 2e^{i\phi} s_+ s_- + 1 \right)^2 - 4e^{2i\phi} c_+^2 c_-^2} \right] \tag{18}$$

and,

$$\lambda_\pm(\pi) = \frac{e^{-i\phi}}{2c_+ c_-} \left[ e^{2i\phi} + 2e^{i\phi} s_+ s_- - 1 \mp \sqrt{\left( e^{2i\phi} + 2e^{i\phi} s_+ s_- + 1 \right)^2 + 4e^{2i\phi} c_+^2 c_-^2} \right] \tag{19}$$

for $E = \pi$. Graphs of (17) for various parameters are shown in Fig. 4.

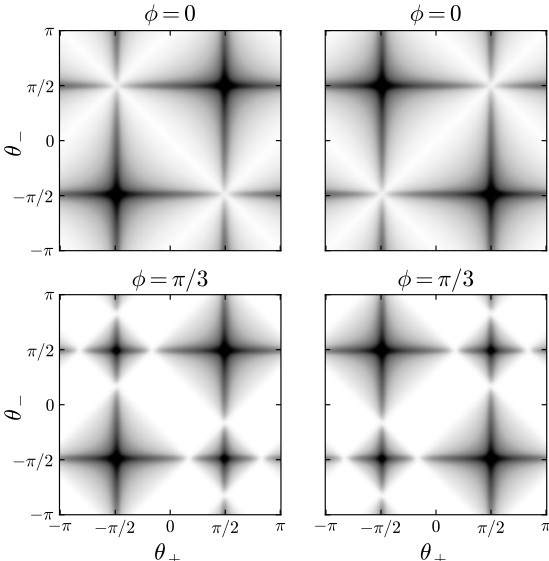

Figure 4: Inverse of the localization length $\Lambda(E) = \xi_E^{-1}$, for $E = 0$ (first column) and $E = \pi$ (second column), and two values of the interaction phase $\phi = 0, \pi/3$. The main effect of the interaction is to increase the area (white regions), in the parameter space $(\theta_+, \theta_-)$ with diverging $\xi$ (unlocalized states). Grey scales from white (zero level) to black ($\Lambda(E) = 6$).

This simple computation shows that the interaction changes the localization properties of the quantum walk, allowing for instance the existence of delocalized states in regions of the parameter space where, without interaction, the states are localized, and vice-versa. These interacting, delocalized states correspond to the binding states, for which the two particles occupy the same position (molecular states). This is just the effect found in the calculation of the localization length, predicting the spreading over the line of the molecular state.

To illustrate this point, we plotted in Fig. 5, the distribution probability of the two particles, for an initial symmetric Bell state (state '0'), and for two strengths of the interaction. The origin separates a zero charge ($x < 0$) region to a one charge ($x > 0$) region, creating at zero interaction, a localized state (Fig. 5, first pannel). For finite interaction, a two particle bound state appears, which may propagate away from the origin. However, the probability distribution along the propagation line $x_1 = x_2$ lost its symmetry with respect to the origin (Fig. 5, second pannel). This is a consequence of the edge state at the origin, which do not disappear with the interaction in this range of parameters. Another behavior is observed without interface for $\phi = 3\pi/4$ (Fig. 5, third pannel), a localized molecular state appears together with enhanced repulsion for an initial symmetric state, more reminiscent of an antisymmetric state. The opposite situation, a free localized state becoming extended in the presence of interaction, is found for exemple in the case of an antisymmetric initial state (last pannel of Fig. 5); propagation in the $x_1, x_2 < 0$ region, inhibited for $\phi = 0$, is reestablished for $\phi = 3\pi/4$.

In order to delimitate the regions in parameter space possessing localized states, we computed the probability to stay at the origin for one or two particles:

$$P(t) = \sum_{s_1 x_2 s_2} |\langle 0 s_1 x_2 s_2 | \psi(t) \rangle|^2, \tag{20}$$

$$P_0(t) = \sum_{s_1 s_2} |\langle 0 s_1 0 s_2 | \psi(t) \rangle|^2, \tag{21}$$

respectively. If these probabilities remain finite, decreasing at a rate slower than $1/N$, after $N$ time steps, we say that localized states exists. This is a qualitative definition that seems to

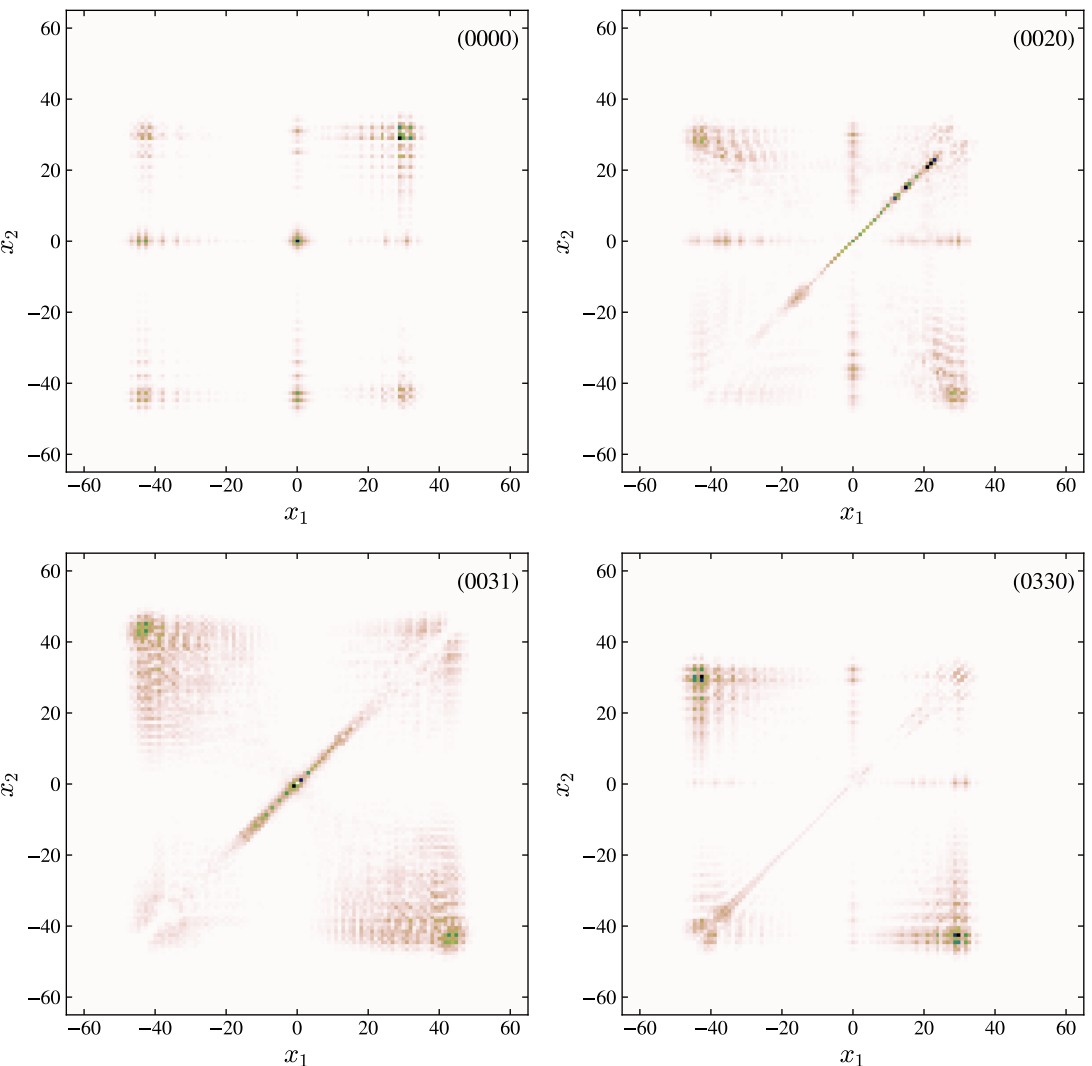

Figure 5: Two particle probability distribution (after 60 steps), with initial state Bell '0' and left charge '0' and right charge '1'. In the non-interacting case there is a localized state at the origin (left, code (0000)); in the presence of interaction, $\phi = \pi/2$, a binding state of the two particle becomes itinerant (right, code (0020)); for a weak interaction strength the localized state persists. In the trivial interacting system, a localized state appears (0031), for $\phi = 3\pi/4$, while for the antisymmetric state (0330) is delocalized, in spite of the interface.

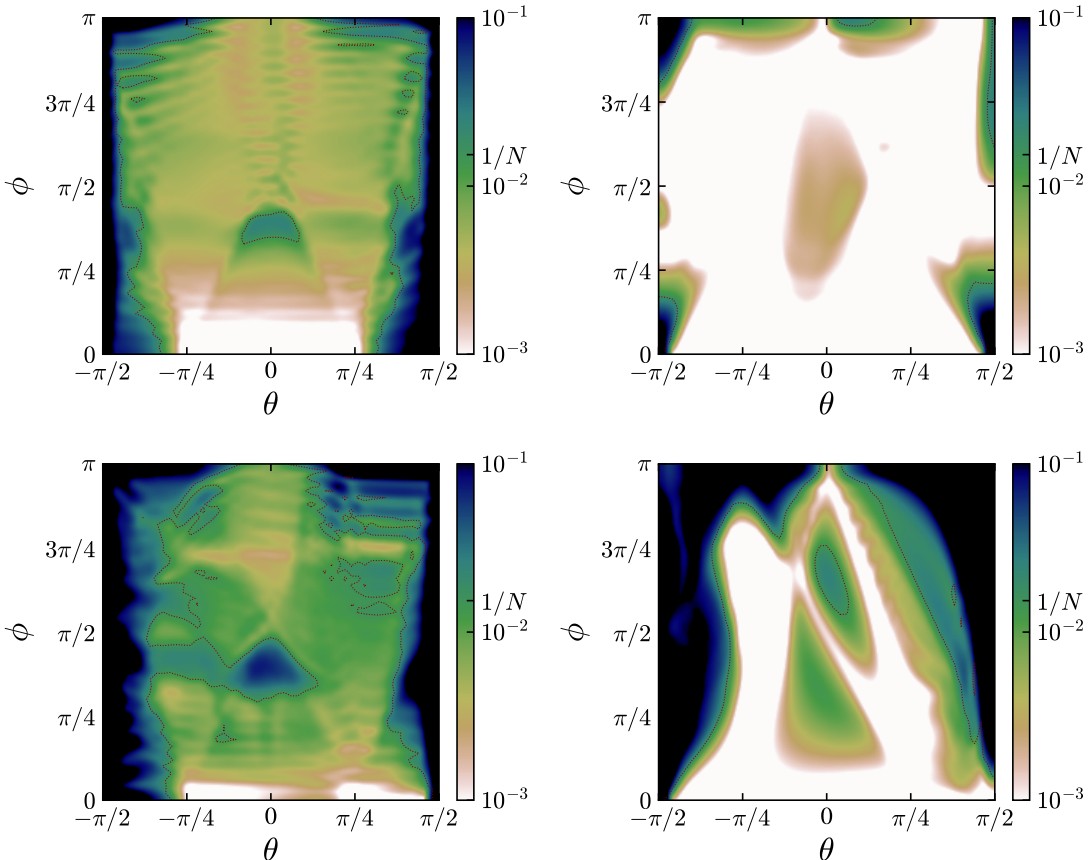

Figure 6: Probability of one particle to stay at the origin for different parameters $(\theta, \phi)$, in a logarithmic scale. The level $1/N$ defines the frontier of the localized states ($N = 65$). The interface separates a left region in a '0' phase (first row) or '1' phase (second row), and a right region, which is in phase $-1$ for $\theta < \pi/4$, 1 for $\theta > \pi/4$, and 0 in between. The first column corresponds to an initial '0' Bell state (symmetric), while the second column corresponds to a '3' Bell state (antisymmetric). One observes (first row) the emergence of a localized region, induced by the interaction $\phi$, for the '00' symmetric case that disappears for the antisymmetric '03' case, for which almost all parameter space corresponds to delocalized states. A similar qualitative behavior is found for an interface with charge '1' on the left (second row).

be precise enough to detect a range of parameter values corresponding to localization at the origin. In Fig. 6 we fixed two values of the left rotation angle $\theta_L$, and two initial states $\phi_+$ and $\psi_-$, and varied the right angle $\theta$ and interaction phase $\phi$. We plot $P(t)$ as a function of $(\theta, \phi)$ for the two Bell states, one symmetric and the other one antisymmetric (columns), and two types of interfaces, with charge zero or one on the left region (rows). Note the complex pattern of the localized-delocalized regions with the emergence of localization in areas of the parameter space, and their intricate boundary, especially in the case where the left region $x < 0$ possesses a $c = 1$ topological charge.

One important characteristic of the quantum behavior of the system is related to the growth of the entanglement entropy. As a measure of the entanglement we use the von Neumann entropy of the position of particle one degree of freedom $S_x$, obtained by tracing out the remaining quantum degrees of freedom (second particle position, and spin state),

$$S_x(t) = -\text{Tr}\,\rho_x(t)\log\rho_x(t), \ \rho_x(t) = \text{Tr}_{\bar{x}}\rho_{x,\bar{x}}(t), \tag{22}$$

where $\rho = \rho_{x,\bar{x}}$ is the density matrix of the total system, and the partial trace $\text{Tr}_{\bar{x}}$, is over the

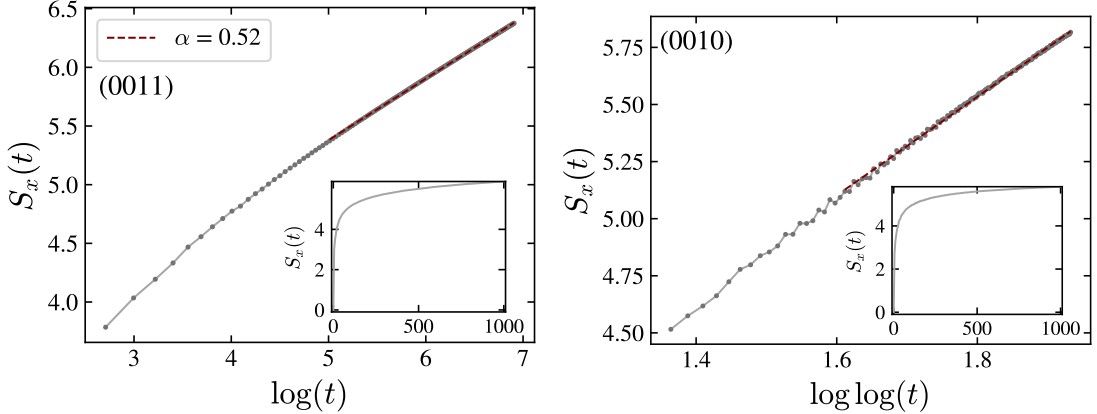

Figure 7: Entanglement entropy of particle one position as a function of time for case (0011), without topological interface, and case (0010), with a '01' interface. In the first case (left pannel), there is a binding state for the two particles, and the entropy grows following a log law, with an exponent $\alpha \approx 0.54$; in contrast, for the second case (right pannel), a localized state is present at the origin, and the entropy growth is only $\log\log$. The insets show the unscaled graph.

Hilbert space labeled by the set of quantum numbers $\bar{x} = \{c_1, x_2, c_2\}$, the complementary set of $\{x = x_1\}$.

To analyse the entropy growth, we assume that for enough long times, there exists a probability distribution $p_S = p_S(x, t)$ such that the von Neumann entropy can be approached by the expression,

$$S_x(t) \approx -\int_{-\infty}^{\infty} \mathrm{d}x\, p_S(x, t) \log p_S(x, t), \tag{23}$$

with the normalization,

$$\int_{-\infty}^{\infty} \mathrm{d}x\, p_S(x, t) = 1. \tag{24}$$

In the delocalized state, we consider that the growth of the entanglement can be described by a self-similar form:

$$p_S(x, t) = \frac{1}{t^\alpha} P_S\left(\frac{x}{t^\alpha}\right), \tag{25}$$

with the self-similar variable $X = x/t^\alpha$, characterized by a single exponent $\alpha$. This form guarantees that the normalization condition is automatically satisfied if $P_S$ is itself normalized. Substituting into (23), and using (24), we obtain

$$S_x(t) = S_0 + \alpha \log t, \tag{26}$$

where

$$S_0 = -\int_{-\infty}^{\infty} \mathrm{d}X\, P_S(X) \log P_S(X).$$

The exponent depends in general on the interaction strength $\alpha = \alpha(\phi)$, vanishing for $\phi = 0$. The self-similar form (25) is borrowed from the one particle quantum walk, for which the ballistic law $xt$ applies [2, 35]; this is certainly appropriated when the probability distribution is dominated by the motion of the bounded state of the two particles, and then we have an effective one particle walk, or in the other limit, when there is repulsion (as in the antisymmetric case) and the two particles are almost independent. The fact that the characteristic exponent

varies with the interaction can be related to the leak of one particle position probability in correlations with the other degrees of freedom: the effective motion of one particle is no more ballistic.

In the localized state, for times of the order of $t(x) \sim e^{x/\xi}$ for $x \gg \xi$ (where $\xi$ is the localization length), the entanglement probability $p_S(x,t) \to P_S(x/\log t)$ tends towards an almost stationary distribution. This leads to a very slow, double logarithmic growing law,

$$S_x(t) \sim \log(\xi \log t), \tag{27}$$

difficult to test numerically.

In Fig. 7 we represented the entanglement entropy for a delocalized two particle bounded state (left, labeled '0011') and for a localized state (right, labeled '0010'), for the initial Bell state '0', illustrating the two growing modes.

## 4 Discussion and conclusions

We investigated the localization and entanglement properties of a system of two interacting particles executing a topological quantum walk. The behavior of a two particle quantum walk cannot be straightforwardly deduced from the one particle case, even without interaction [18, 26], new effects arise related to the symmetry of the initial state (which is preserved by the interaction) and the wrapping of the energy bands.

The presence of edge states at the interface of two regions with different topology, and the appearance of interacting molecular states, further modifies the properties of the system. In fact, the evolution of the system depends on the type of the interface; an interface between charge $-1$ and $0$ regions, or $1$ and $0$ are not equivalent. Furthermore, the overlapping of the initial state with the edge state (for example, putting the particles on the left or on the right of the interface), will affect the details of the system's evolution. However, the basic properties, like the localization and binding effects, are robust, they do not depend on the details of the initial condition.

More specifically, we showed that new localization-delocalization transitions were possible, depending on the interaction phase and the symmetry of the quantum states. Antisymmetric states favor delocalization of the walk, and the unbinding of the molecule. Symmetric states, on the contrary, preserve the edge states and the molecular binding. For the particular value $\phi = \pi$, strongly localized states are observed. For intermediate values, in the range $\phi = (\pi/3, \pi/2)$, localized states in the free delocalized region appear, for symmetric states, but also for antisymmetric states depending on the topology.

Therefore, the effect of the interaction is that it can induce localized states, absent for free particles, in the trivial topological case. This effect is present, for different values of the interaction strength, independently of the initial state. The molecular states, which can propagate along the line, compete with the localized state, which tends to stick the particles at the interface. Moreover, depending on the phase of the interaction, a delocalization is possible in the presence of the interface. Indeed, the interaction can also destroy the edge states to allow the propagation of the information between regions with different topology. Therefore, the interaction can be used as a control parameter of the information transport.

Interaction not only contributes to modify the localization properties of the quantum walk, it is the physical mechanism of entanglement generation. We measured the entanglement entropy of one particle and found that it steadily increases over time [12, 36]. However, at variance with other interacting systems, its growth in not simply proportional to time, but shows two different regimes according to the localization of the walk. For delocalized states

the entropy growth logarithmically, while for localized states it growth double logarithmically. We showed some numerical evidence of these different growth laws.

In conclusion, using a quantum walk we studied various basic mechanisms, such as the interplay of topology and interaction, or the relation between entanglement and interaction. These effects also have interesting analogies with condensed matter systems. Indeed, quantum walks may be used to simulate material systems [37], or can be related to specific condensed matter systems via their effective hamiltonian. Simple quantum walks defined by a set of local unitary operations, can be equivalent to complex nonlocal effective quantum systems. We observed for instance, the growing of the entanglement entropy associated with long range correlations induced by a local interaction rule; or the appearance of localized states without disorder, induced by the interaction in a background trivial topology.

## Acknowledgements

We thanks Simon Lucchesini who wrote a first version of the numerical code. We benefited from discussions with Giuseppe Di Molfetta, Laurent Raymond and Luis Foa Torres. Part of this work was funded by the Université de Toulon.

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
