# Peer review of "Entanglement and interaction in a topological quantum walk"

_SciPost Physics, doi:SciPost Phys. 5, 019 (2018)_

## Round 1 · Referee Report · Anonymous (Referee 1) · 2018-6-10

Strengths

1) Interesting discussion about edge states in a topological quantum walk. 2) Detailed study of localization in relation to interaction and topology.

Weaknesses

1) The effect of the symmetry of the initial state is not clear to me, since the interaction is spin-independent. 2) The authors have not been careful with notations, specially in the figure captions.

Report

The authors study the quantum walk of two interacting particles on a line with an interface separating two topologically distinct regions, and analyze the localization and entanglement properties of the edge state as a function of the interaction strength and topology interface.

In my opinion, the paper is interesting. However, there is a couple of questions that should be addressed before I can recommend publication.

  • The interaction operator Eq. (6) is spin independent (this equation can in fact be written in terms of position projectors only). Then, it is not intuitive how the dynamics can depend on whether the initial state is spin-symmetric or antisymmetric, while keeping the rest of parameters unchanged. The authors should clarify this point in connection with Figs. (5) and (6).

  • The authors should give a justification for Eq. (24).

The authors have not been very careful about the notations introduced in the paper, specially in the figure captions. Here are some comments:

  • Regarding Fig. 2: What represent black dots in the upper panel? Are there quasienergies that do not correspond to eigenvalues of P? The discussion about this figure is confusing.

  • The authors introduce a position dependence of \theta_+(x) characterized by \theta_L and \theta_R. Is this dependence kept throughout the rest of the paper? For example, in Fig. 2 they write specific values for \theta_+= and \theta_.

  • Same for Fig. 4. Also, the definition of \Lambda in the caption does not agree with Eq. (16).

  • In Fig. 6: What is \theta? The caption of this figure has to be carefully checked. Is this figure calculated for a specific value of N?

  • How is \omega appearing after Eq. (15) related to E?

  • The introduction of the reduced density matrix \rho_x(t) in Eq. (21) is confusing. What does it mean \bar{x}=(c_1,x_2,c_2)?

  • The use of P_E(x,log t) before Eq. (26) is misleading.

  • The notation (c,b,g,i) used to label different sets of parameters is not transparent. I recommend using a more explicit notation.

Requested changes

1) Clarify the role of symmetry of the initial state. 2) Check notations.

---

## Round 2 · Referee Report · Anonymous · 2018-7-30

Strengths
1) Interesting discussion about edge states in a topological quantum walk.
2) Detailed study of localization in relation to interaction and topology.
Report
The authors have successfully addressed the comments and criticisms in my previous report. I recommend publication.

---

## Round 2 · Referee Report · Anonymous · 2018-8-21

Strengths
1- Clearly written.
2- Mathematical formalism is intuitive and clear
Weaknesses
1- n/a
Report
Dear Editor,
In this paper Verga and Elias present an analytical study of a quantum walk comprising two walkers on a linear graph where an interface is introduced that separates the region into distinct topological areas.
The study of topological phases has become somewhat of a hot topic in the field of quantum walks in particular, making this study timely.
To study this system the authors present both analytic and numeric results studying localization and entanglement properties and dynamics, and consider the effects of different symmetries of the input state.
The basic model upon which their study is based (starting with Eq. 1) is intuitive and reasonable. And the measures for localization and entanglement are also well justified.
The authors found that new (de)localization transitions are possible under this model, as a function of the phase interactions and input state symmetries. Of particular interest was the observation that information can ‘flow’ between topologically distinct regions, which might have interesting consequences beyond the scope of this manuscript. The authors found some interesting connections between the temporal dynamics of their chosen entanglement measure (von Neuman entropy) and the walkers’ localization.
I found this manuscript interesting (from the perspective of a reader in the field of quantum walks) and some of the authors’ observations may extend beyond the relatively limited scope of quantum random walks. Although not groundbreaking, I feel this manuscript is a useful addition to the literature in this field. Some of the observations made by the authors could stimulate interesting follow-up work.
For these reasons I believe this manuscript should be published and I recommend publication.
Requested changes
None requested

---

## Round 2 · Author Response

Response to the referee Report 1:
We added two paragraphs to address the questions about the symmetry of the interaction and the self-similar form of the probability in (24) (now equation 25).
We revised the notation, figure captions and errors.

---

## Round 2 · List of Changes

* (p4. before section 3) Dependence of the dynamics on the symmetry of the initial state:
> In summary, the model consists in two particles moving in a one dimensional lattice with an interface at the origin; their evolution is ensured by a one time step operator $U$, composed of a coin (defined with different rotation angles in the two regions), which couples to the shift operator for each particle, and a local on-site interaction between the two particles. Although the interaction is spin independent, the global unitary operator cannot be splitted into a spin dependent part and a position dependent part. Coin and shift entangle spin and position in the one particle sector, and the interaction the positions in the two particle sector; as a consequence, the interaction can modify differently the evolution of a given initial condition depending on its symmetry: symmetric and antisymmetric Bell states \eqref{e:bell}. In other words, $U = V \,(U_0 \otimes U_0)$ entangles spin and position ($U_0$) of each particle and the two particle positions ($V$), resulting in entanglement of all degrees of freedom labeled by the basis vectors $|x_1s_1x_2s_2\rangle$ of the Hilbert space.
* (p.11 after the equation for $S_0$) About equation (24) (now 25); we also changed the notation $p_E$ to $p_S$ to avoid confusion with the energy $E$:
> The self-similar form \eqref{e:pe} is borrowed from the one particle quantum walk, for which the ballistic law \(x \sim t\) applies \cite{Nayak-2000qv,Venegas-Andraca-2012zr}; this is certainly appropriated when the probability distribution is dominated by the motion of the bounded state of the two particles, and then we have an effective one particle walk, or in the other limit, when there is repulsion (as in the antisymmetric case) and the two particles are almost independent. The fact that the characteristic exponent varies with the interaction can be related to the leak of one particle position probability in correlations with the other degrees of freedom: the effective motion of one particle is no more ballistic.
### Notation, figure captions, errors, etc.
In response to the inconsistencies in the notation and mistakes in the formulas or figure captions, we addressed the following points:
1. We changed the text and the figure caption of Fig. 2 to improve its presentation.
2. We changed the text about the definition of the left and right regions, and added when necessary reference to $\theta$.
3. Corrected (definition of $\Lambda$ in caption 6.) Modification of the text to improve the reading.
4. We changed $\omega$ by $E$ (we used omega in the by hand computation with sympy...)
5. We changed the text before (22) (former 21) to clarify the notation and the definition of the reduced density matrix
6. Corrected: $P_S(x/\log t)$
7. We completed the explanation of the code (cbgi) by an example, and slightly changed the notation of the labels.
### Other modifications
We added a reference to the paper of Bisio et al. who computed the spectrum of a similar operator $U$.
We added references to one particle quantum walks in relation with the discussion of the self-similar probability form.
We corrected errors in the text and expanded some comments on the figures in the main text.

---

## Editorial Decision

published